# Estimates of the cost to build a stand-alone environmental surveillance system for typhoid in low- and middle-income countries

**Brittany Hagedorn**[1]*, **Nicolette A. Zhou**[2], **Christine S. Fagnant-Sperati**[2], **Jeffry H. Shirai**[2], **Jillian Gauld**[1], **Yuke Wang**[3], **David S. Boyle**[4], **John Scott Meschke**[2]

1 Institute for Disease Modeling, Bill & Melinda Gates Foundation, Seattle, WA, United States of America,
2 Environmental and Occupational Health, University of Washington, Seattle, WA, United States of America,
3 Center of Global Safe Water, Sanitation, and Hygiene in the Hubert Department of Global Health, Emory University, Atlanta, GA, United States of America, 4 Diagnostics Program, PATH, Seattle, WA, United States of America

* Brittany.Hagedorn@GatesFoundation.org

**Data Availability Statement:** All input values used in this analysis are provided in the supplemental materials. The unaggregated cost survey data are

## Abstract

The typhoid conjugate vaccine is a safe and effective method for preventing Salmonella enterica serovar Typhi (typhoid) and the WHO's guidance supports its use in locations with ongoing transmission. However, many countries lack a robust clinical surveillance system, making it challenging to determine where to use the vaccine. Environmental surveillance is one alternative approach to identify ongoing transmission, but the cost to implement such a strategy is previously unknown. This paper estimated the cost of setting up and operating an environmental surveillance program for thirteen protocols that are in development, including thirteen cost components and twenty-seven pieces of equipment. Unit costs were obtained from research labs involved in protocol development and equipment information was obtained from manufacturers and the expert opinion of individuals in participating labs. We used Monte Carlo simulations to estimate the costs and the input parameters were modeled as distributions to incorporate the uncertainty. Total costs per sample including setup, over-head, and operational costs, range from $357–794 at a scale of 25 sites to $116–532 at 125 sites. Operational costs (ongoing expenditures) range from $218–584 per sample at a scale of 25 sites to $74–421 at 125 sites. Eleven of the thirteen protocols have operational costs below $200, at this higher scale. Protocols with higher up-front equipment costs benefit more from scale efficiencies and sensitivity analyses show that laboratory labor, processes, and consumables are the primary drivers of uncertainty. At scale, environmental surveillance for typhoid may be affordable (depending on the protocol, scale, and geographic context), though cost will need to be considered alongside future evaluations of test sensitivity. Opportunities to leverage existing infrastructure and multi-disease platforms may be necessary to further reduce costs.

not available due to the sensitivity of individual laboratories' costs being identifiable due to the small sample size.

**Funding:** The authors received no specific funding for this work.

**Competing interests:** The authors have declared that no competing interests exist.

## Introduction

The typhoid conjugate vaccine (TCV) was recommended for use by the World Health Organization (WHO) in guidance that they issued in March of 2018 for populations at risk for typhoid fever [1]. Gavi, the Vaccine Alliance reports that the TCV was pre-qualified by the WHO in December 2018 [2], making it available to qualifying low-income countries and introductions began the following year, with Pakistan being the first country in the world to include it in their routine immunization schedule.

Stanaway et al. [3] and Marchello et al. [4] estimate the global burden for typhoid, but these are based on very limited data, since most Gavi-qualified countries have few, if any, sites that conduct routine clinical surveillance for typhoid fever. With many competing health priorities, generating evidence of the true burden of disease and thus at-risk populations, is critical for countries trying to decide when and if to fund the introduction of TCV. The information available to assess sub-national burden is even less, making it difficult to consider a targeted vaccination strategy and to assess the impact of a TCV roll-out. Because of the limited feasibility and high cost of long-term expansion of clinical surveillance in these settings, environmental surveillance (ES) is being evaluated as an alternative for continued monitoring for the presence of typhoid circulation, for example by Murphy and Verani [5] in Kenya. Jeon et al. [6] and Wang et al. [7] have proposed methods for selecting sites for use in vaccine decisions, although protocols are still in development.

Tebbens [8] examined the total cost of the global polio laboratory network (including both environmental and case surveillance) and concluded that true total costs were higher than budgeted for, while Hagedorn et al. [9] found that there were economies of scale for surveillance programs. Given this, cost estimates for a future typhoid ES system must consider costs of implementation not for a single sampling site, but when expanded to cover large geographic regions and higher volumes. Additionally, Geng et al. [10] found in their costing studies of healthcare systems in low- and middle-income countries (LMIC) that delivery and overhead costs may account for 31–87% of site-level costs, highlighting the need to factor them into estimates of future surveillance costs.

Due to the ongoing nature of the development and field testing of ES protocols for typhoid, there remains a substantial amount of variation in sample processing and laboratory methods, which needs to be accounted for. A survey of laboratories developing typhoid environmental surveillance protocols showed that protocols were highly variable with regards to sample collection method, concentration, enrichment steps, and analysis methods; the thirteen identified potential ES protocols, from the seven laboratories participating in the study, are summarized in Table 1.

The methods summarized in Table 1 may have differing sensitivity and specificity, cost, and operational scalability, which would all impact how successfully they can be implemented in a routine monitoring system. This is valuable to inform accurate budgeting, but more importantly, it is needed in order to guide realistic target-setting for 'acceptable' costs per sample as part of target product profiles (TPPs) and for informing discussions on how much of the total cost an LMIC will be able to fund themselves, and thus how much donor support may be needed.

This paper explores the drivers of costs for different protocols and quantifies the differences in both per-sample cost and the efficiency of scaling up to higher numbers of samples. By estimating the total costs across multiple potential protocols, any comparisons between them will then be on equal footing, since the estimates will all include the major cost drivers and have comparable assumptions, thus informing trade-off discussions and future research and development strategies. These estimates are intended for use in strategic planning, but due to the variation between countries in their labor markets, opportunity for horizontal integration with other surveillance programs, and financial costs, they should not be used for budgeting purposes.

**Table 1. Methods employed in each protocol.** Protocols (named in the rows) are composed of multiple methods (names in columns). X intersections represent methods that are included in each protocol and are relevant for costing purposes. Each method (in the columns) requires different resources, which impact the cost of sample collection and processing.

| Protocol | Concentration Method | | | | | | | | | Detection Method | | |
| --- | --- | --- | --- | --- | --- | --- | --- | --- | --- | --- | --- | --- |
| | Filter cartridge (FC) | Grab enrichment (GE) | Moore swab (MS) | Dead-end ultra-filtration (DEUF) | Differential Centrifugation (DC) | Membrane filtration (MF) | Tangential flow ultra-filtration (TFUF) | Enrichment (E) | DNA extraction (DNA) | Culture (C) | qPCR | qPCR TAC |
| FC-qPCR | X | | | | | | | | X | | X | |
| FC-qPCR(TAC) | X | | | | | | | | | | | X |
| GE-E-qPCR | | X | | | | | | X | X | | X | |
| MS-E-qPCR | | | X | | | | | X | X | | X | |
| DEUF-E-qPCR | | | | X | | | | X | X | | X | |
| DC-E-C-qPCR | | | | | X | | | X | X | X | X | |
| DC-qPCR | | | | | X | | | | X | | X | |
| MF-qPCR | | | | | | X | | | X | | X | |
| MF-E-qPCR | | | | | | X | | X | X | | X | |
| MF-E-C-qPCR | | | | | | X | | X | X | X | X | |
| TFUF-qPCR | | | | | | | X | | X | | X | |
| TFUF-E-qPCR | | | | | | | X | X | X | | X | |
| TFUF-E-C-qPCR | | | | | | | X | X | X | X | X | |

FC = filter cartridge. GE = grab enrichment. MF = membrane filtration. MS = Moore swab. TFUF = tangential flow ultrafiltration.

## Methods

The cost of a full-scale typhoid ES laboratory system is currently unknown, since it has not yet been built, but there is a need to understand the magnitude of the total costs in order to begin planning for funding and implementation. Here, we made the first estimate of the total and per-sample costs for thirteen protocols that are currently in development or use (Table 1, Table A in S1 Text, Table B in S1 Text, Table C in S1 Text, Table D in S1 Text, Table E in S1 Text). We estimated the full cost to run a laboratory dedicated only to typhoid ES, which resulted in conservative (*i.e.*, most expensive) estimates. We took a programmatic perspective and included financial costs only. All values are reported in 2019 U.S. dollars.

### Data collection and assumptions

We made use of unit cost surveys (detailed in S2 Text) conducted by the Environmental and Occupational Health Microbiology Laboratory at the authors' institute, as part of their evaluation of typhoid ES protocols, and when data were not available, estimates were informed by expert opinion provided by staff in participating laboratories. Survey respondents included seven labs conducting a total of thirteen protocols in Bangladesh, India, Kenya, Malawi, Nepal, and Pakistan. Costs were reported in the survey in US dollars based on the laboratories' real expenditures. Because survey responses included details on specific lab research budgets, we aggregated survey results so that individual labs cannot be identified. The first quartile, median, and third quartile were used to parameterize the uncertainty in the distribution of unit cost values.

The total cost model was a generalized exploration of total costs and unit costs at scale. It did not consider localized effects specific to countries, such as import duties on equipment, fuel rate variations, or tax rates on various components of the input costs.

The cost model included components for labor, equipment, maintenance, consumables, depreciation, and overhead. Each of these had multiple inputs as described below and were calculated using an ingredients-based approach. Surveys reported per-sample unit and consumables costs (*e.g.*, reagents), required equipment, labor time per sample, and samples collected per day per team. Expert opinion was provided by personnel adept in conducting the field and laboratory methods, and informed the list of required equipment, equipment capacities, labor time per sample, laboratory batch size, technicians per team, and samples collected per day per team. Available commercial pricing in the United States informed consumables and equipment costs where there were gaps in the survey data; this is appropriate because equipment is regularly purchased and then shipped from US or European markets. A comprehensive listing of all equipment requirements (Table A in S1 Text), unit costs (Table B in S1 Text, Table C in S1 Text), equipment capacities and lifespans (Table D in S1 Text), labor time per sample (Table E in S1 Text), laboratory batch size (Table E in S1 Text), technicians per team (Table E in S1 Text), and samples collected per day per team (Table E in S1 Text) is in the supplement.

The input value that has the greatest level of variation (from the data reported in the cost survey) is the daily pay rate for technicians. This is because of the widely varying labor markets across low- and middle-income countries, with pay rates depending on the local economy and inflation, alternative job opportunities, and the influence of international employers on the local job market. Reflecting this reality, the cost surveys ranged from $3.50 to $40.00 per day for technicians (Table C in S1 Text). These are used to parametrize the level of uncertainty in labor costs in the model and thus impact the confidence intervals reported in the results (details in Table A in S3 Text).

For the purposes of this paper, the term 'method' refers to a single step in the protocol, such as DNA extraction, and the term 'protocol' refers to the combination of methods that make up the complete series of steps of collecting and processing of a sample.

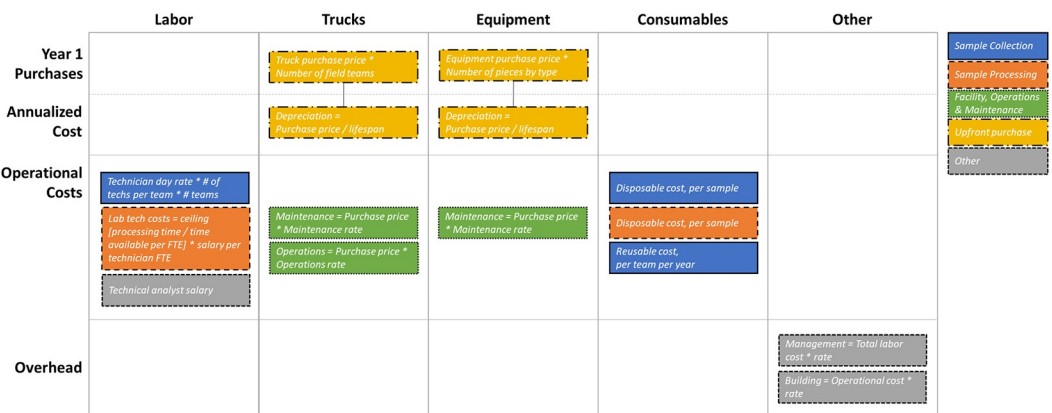

**Fig 1. Cost model components, categorized by type and color-coded by function.** Detailed descriptions and formulas can be found in Table A in S3 Text. Operational costs include labor, truck operations and maintenance, equipment maintenance, and consumables.

### Model structure

We calculated the year-one investment required for capital expenses (*e.g.*, new equipment purchases) and the recurrent operational costs separately; these sum to the total cost. The model components are shown in Fig 1 and the formulaic structure of the model is detailed in S4 Text. For each cost component, whenever the number of samples being processed exceeded capacity, we incremented the number of processing units until it was adequate. For example, if a particular piece of equipment had a weekly capacity of 100 samples, then to process 101 samples, we assumed the purchase of a second piece of equipment and incremented the cost. The same was true for the number of collection teams and laboratory FTEs; when the number of samples exceeded the capacity of the staff, there was incremental cost to bring in additional staff to meet these needs.

For this modeling exercise, we assumed bi-weekly sampling at each collection site (*i.e.*, 26 samples per site per year), that staff typically work 46 weeks per year (*i.e.*, a conservative assumption of six weeks of absence for holidays, vacation, sick, and other absenteeism time), 5 days per week, and that laboratory staff are productive during 80% of their working hours (*i.e.*, the remainder is spent on non-productivity tasks such as trainings, planned maintenance, supply management, etc.). This availability is applied to calculate the appropriate number of laboratory staff required to be hired. These assumptions are based on previous studies by the WHO [11] from Indonesia, Mozambique and Uganda, which showed a range of days off and productive hours. We expect that sample collection staff are flexible and could be paid on a daily-rate basis. We assumed that laboratory staff were hired full time, regardless of their utilization level, due to the technical skills required. Management and building overhead costs were estimated with a traditional approach, multiplying the appropriate percentage to other component costs (see S4 Text).

The model was built in R Studio version 1.3.959, using the R statistical programming language version 3.6.3. Code is available on GitHub upon request.

### Model application

The cost model simulated 1,000 trials for each protocol of varying collection and laboratory sample processing methods. Each unit cost, time duration, and percentage value were sampled for each trial as part of a Monte Carlo simulation from a truncated normal distribution. For

each distribution, we used the survey-reported variation in values; where there were too few data points available, we assumed a standard deviation of twenty percent, in alignment with the variance found in better-sampled cost distributions from the study. For example, for each of the 27 potential pieces of equipment there is an equipment-piece-specific maintenance rate pulled from the appropriate distribution, each with the same expected value but varying because of the stochastic draws; these were multiplied by the base purchase prices (which were also sampled), respectively.

To calculate the total cost per sample processed and assess the impact of scale on the per-sample cost, we assumed a constant use of a single central laboratory facility, but varied the number of samples collected, assuming bi-weekly collection (*i.e.*, 26 samples per site per year) and varying the number of sampling sites from 25 to 125, to represent geographic coverage from a small region to a country-wide strategy. Increases in the number of sites impacted the number of collection teams and trucks required to reach these sites; if additional equipment or laboratory staff were required to process the increased volume of samples, those were also included in the cost-per sample calculations. These requirements are summarized in S2 Text.

Total annualized costs and operational costs are the two primary results. Operational costs included labor, truck operations and maintenance, equipment maintenance, and consumables (including both reusables and disposables); these are expenses that would need to be budgeted on an ongoing basis. Total annualized costs also included the purchase price for trucks and equipment in the form of depreciation, plus management and building overhead expenses. Trucks and equipment categories are calculated separately due to their different cost structures.

Given the uncertainty inherent in the ongoing development of an ES protocol for routine use, we also conducted a sensitivity analysis to assess the impact of factors on either process or unit costs. This sensitivity analysis examined the impact of a 20% change in values for each of the following factors: samples collected per team per day, equipment capacity maximums, laboratory processing time per batch, depreciation lifespan for equipment and trucks, management and building overhead cost rates, collection labor pay rates, equipment and truck purchase costs, equipment and truck maintenance and operational costs, and consumables costs per team and sample.

## Results

### Cost per sample by protocol

The modeled costs for each protocol are shown in Fig 2 and the values are quantified in Table A in S3 Text. Annualized total costs represent the fully loaded expense of setting up, operating and maintaining the surveillance system. Operational costs are a subset of those, representing the ongoing direct expense of running the system, thus they are always lower than the annualized values. The expected average cost per sample is shown by the size of the bar and varies by protocol, with some being much higher than others. However, these results are for cost only and do not account for the sensitivity of the protocol, which could result in expensive protocols being more cost-effective.

Since the magnitude of a potential surveillance system is unknown, we calculate the per-sample cost at varying levels of sample numbers and assess the impact of economies of scale on the unit cost per sample. Most of the cost savings on a per-sample basis happen with relatively modest increases in sample numbers. It is important to note that the relative cost of protocols does not remain constant with scale; for example, FC-qPCR benefits from scale faster than DEUF-E-qPCR, while the MS-E-qPCR protocol is consistently a lower-cost protocol per sample.

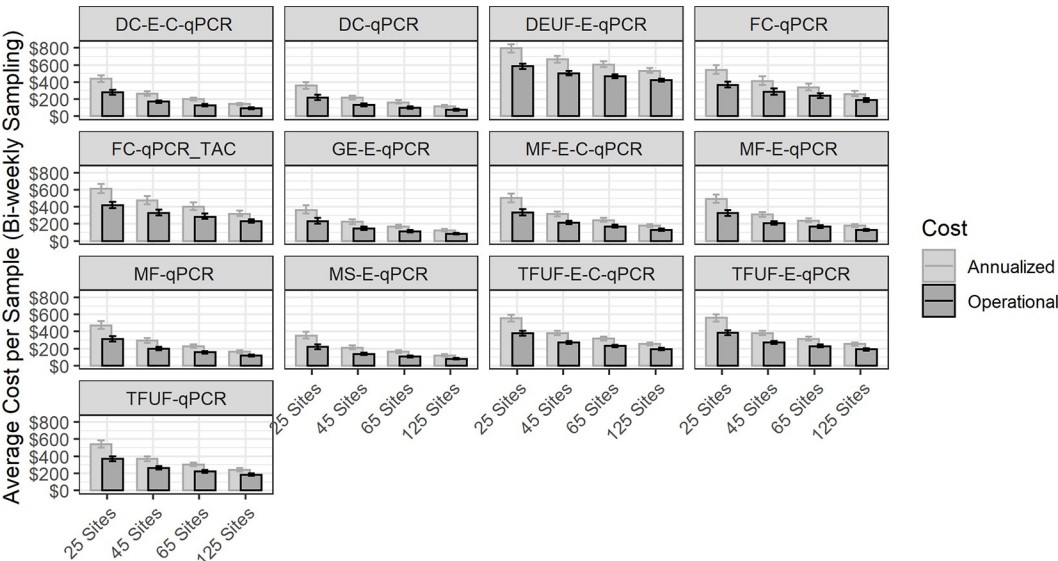

**Fig 2. Total annualized cost (inclusive of all costs, including overhead, buildings, and management) and operational costs (running costs only) for an example number of twenty-five, forty-five, sixty-five, and a hundred and twenty-five sites, sampled bi-weekly.** All values in 2019 US dollars. Error bars represent the 25th and 75th percentile of simulated results. DC, differential centrifugation; DEUF, dead end ultrafiltration; FC, filter cartridge; GE; grab enrichment; MF, membrane filtration; MS, Moore swab; TFUF; tangential flow ultrafiltration.

Total annualized cost and operational cost per sample for each protocol are reported in Table A in S3 Text.

The total cost is the aggregation of six components: labor, equipment, trucks, consumables, management overhead, and the laboratory building (Fig 1). Each of these varies at a different rate and thus benefit from economies of scale to varying degrees. For example, consumables consistently rise as a fraction of the total cost as scale increases because the unit costs per sample are essentially fixed. In comparison, the fraction of the total cost that is spent on equipment and labor decreases with scale as large up-front expenses (*e.g.*, equipment that has a high capacity) are spread across incremental samples. (See Fig 3).

For protocols relying on filter cartridges, dead-end ultrafiltration, and tangential flow ultrafiltration, the cost of consumables becomes the largest component of total cost once a laboratory reaches scale. Other protocols, such as those relying on differential centrifugation, have high upfront equipment investment requirements that continue to be a large component of total cost even at scale. For all protocols, labor is consistently a top-two contributor to costs.

## Sensitivity analysis

Since there is still work in progress to finalize the protocols and implement routine ES, we conducted a sensitivity analysis to identify which of the key cost components have the most impact on the annualized cost per sample. We applied a twenty percent increase to each of the relevant values and the results are shown in Fig 4. The vertical axis categories are ordered according to their aggregate impact across all of the protocols. Two of the three most impactful values are related to the laboratory staff: labor costs (pay rates) and lab processes (how much time a single batch takes to process, which impacts the number of staff needed). Consumable prices also have substantial impact on uncertainty. The equipment- and truck-related costs fall at the bottom of the list, in decreasing order starting from depreciation lifespan, purchase cost, maintenance and operations rates, and equipment capacity.

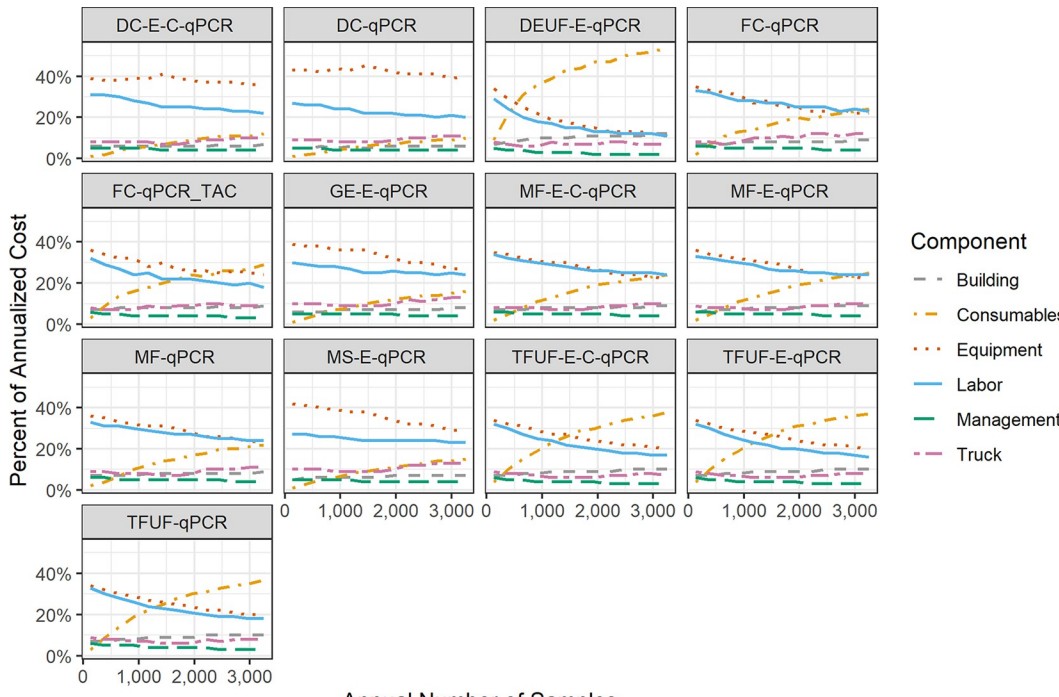

**Fig 3. Breakdown of annualized costs as a percentage of the total for each protocol.** Values are the median of the percentage calculations from 1,000 simulation trials, so the component categories may not sum to 100%. Non-smooth trajectories are indicative of step-changes in cost, for example the purchase of an additional piece of equipment or the need to hire a staff member. Assumes bi-weekly sampling and that the increase in number of samples is due to incremental sampling sites. DC, differential centrifugation; DEUF, dead end ultrafiltration; FC, filter cartridge; GE, grab enrichment; MF, membrane filtration; MS, Moore swab; TFUF, tangential flow ultrafiltration.

## Discussion

The thirteen protocols compared in this study utilize different types of equipment, varied levels of labor intensity, and consumables costs. As a result, the cost per sample can vary widely. At a level of 25 sites, median operational cost per sample ranges from $218 to $584. This decreases with scale; at 125 sites, the range of median operational cost per sample is $74 to $421, with eleven of the thirteen protocols coming in below $200.

In addition to differences between protocols, there is also parameter uncertainty, which we addressed by using a Monte Carlo simulation, with distributions based on the range of unit costs reported by participating labs. These ranges were wide in some cases and the effect of uncertainty on total costs was more pronounced at lower sample volumes.

Economies of scale were a primary driver of the cost per sample, both operational-only and total costs. This was true across all protocols, as up-front costs of labor and equipment are spread over larger numbers of samples. Substantial cost-per-sample reductions were seen up to ~1000 samples per year; after that they continued but at a slower pace due to having achieved most of the efficiencies from large capacity equipment.

However, it is important to note that not all protocols scale in the same way, with some benefiting more than others depending on their mix of semi-fixed labor and equipment costs versus less scalable consumables and collection costs. This has implications for funding mechanisms. In many LMICs, surveillance programs are funded in part through donor support. If countries are eventually expected to pay for the ongoing operations of their surveillance programs, their preferences may be toward protocols that have high up-front investments but

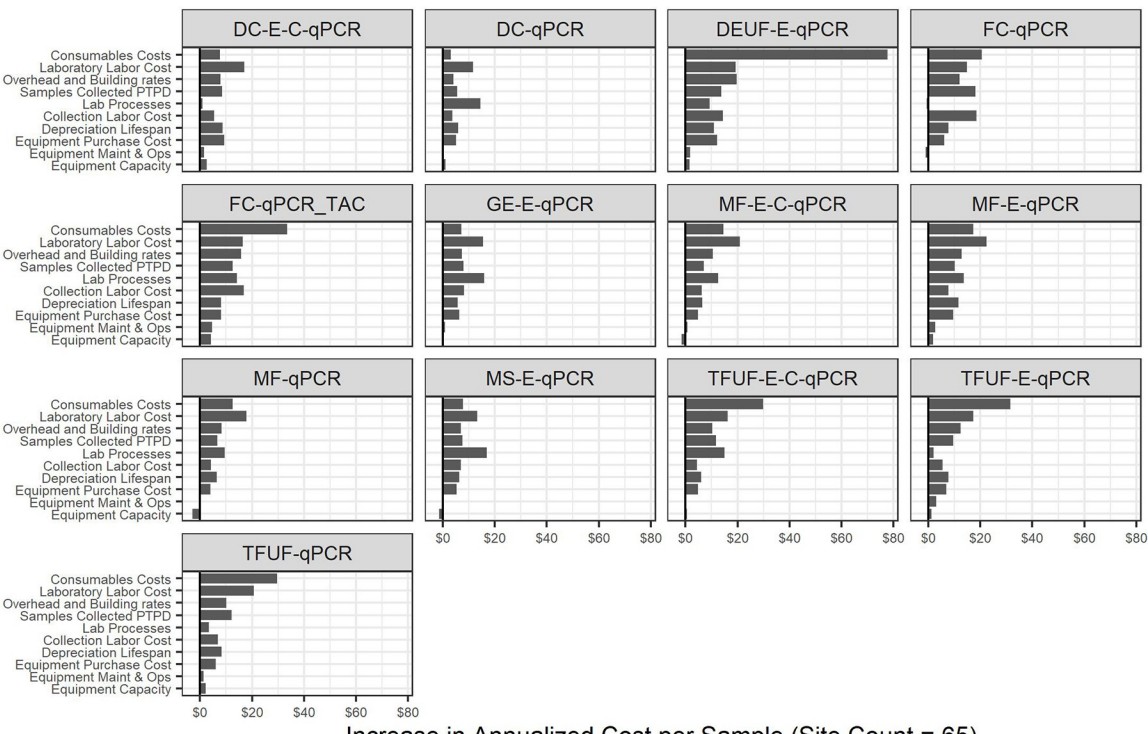

**Fig 4. Sensitivity analysis reflecting the impact of a 20% increase in the relevant baseline value and its impact on the annualized cost per sample.** For example, laboratory labor cost multiplies the salary values by 1.2 to increase the total; depreciation lifespan multiplies the number of years by 0.8, which thus increases costs due to faster replacement. All values in 2019 US dollars. Values represent an example scenario with sixty-five sites and bi-weekly sampling (*i.e.*, 26 samples per year per site). Increases are calculated as the difference between the mean baseline and 20% adjusted scenarios. DEUF = dead end ultrafiltration. FC = filter cartridge. GE = grab enrichment. MF = membrane filtration. MS = Moore swab. TFUF = tangential flow ultrafiltration.

lower ongoing costs, pushing them away from methods with high consumables (such as those using DEUF or TFUF) or laboratory labor costs, for example.

There may also be additional cost savings possible through other means. For example, group purchasing options for laboratory supplies and equipment could result in substantial savings, similar to those at other programs for vaccines and supplies, as shown by DeRoeck et al. [12]). Bulk procurement at discount rates for equipment and reagents is already done by UNICEF's Supply Division, the US Centers for Disease Control and Prevention (CDC), the President's Emergency Plan For AIDS Relief (PEPFAR), and the United States Agency for International Development (USAID); a similar strategy could be employed for ES-related purchasing.

Scale was the largest driver of the cost-per-sample, but we also found via the sensitivity analysis that protocols were particularly sensitive to the cost of laboratory labor, both pay rates and the time to process a sample. Salaries are variable by country, so this implies that the cost of an ES program is substantially dependent on the labor market. This may reduce total cost if the surveillance system is being used in a low-income country where salaries are generally lower, but it does mean that the cost estimates presented here are sensitive to context.

Although the cost of equipment has implications for economies of scale, these costs were less impactful in the sensitivity analysis and the most important factors were the lifespan and the purchase price. These are the two variables required to calculate annual depreciation, which is the largest cost related to equipment and trucks overall. A systematic review

conducted by Diaconu et al. [13] showed that lifespans may be shorter than standard depreciation timelines in LMIC contexts if there are less well-controlled, harsher operating environments, or limited maintenance than we assumed, which would affect our estimates. On the other hand, resource-limited environments may also be forced to utilize laboratory equipment and/or supplies for longer than the depreciation estimates, which too would impact cost projections, so this is a source of uncertainty in the model.

The results presented are based on aggregated cost estimates from a range of geographies, partially obscuring the differences inherent in local economies. Users of these results should consider not only the expected values calculated here, but also how their local input costs compare to those that we used (listed in S1 Text). For example, equipment importation may be substantially more expensive in some locales, making the efficiency of scale more important but also implying that protocols with a high proportion of costs coming from equipment may not be the best option (e.g. DC-qPCR). The same could be true for labor markets, with high local labor costs making some protocols (e.g. MF-E-C-qPCR) less attractive. Decision makers should reference Fig 3 and consider their local context when comparing across protocols.

As protocols are further refined, these estimates will need to be updated with more accurate input values, with a priority on the parameters highlighted through the sensitivity analysis. Equipment is 30–40% of annualized total costs at low volumes and since equipment is a large part of initial investments, details about which manufacturer is approved for use will be critical. For global surveillance programs, tests are often developed and validated for use on certain platforms and thus the manufacturers' decisions about how to set prices (including technical support and maintenance) will have substantial impact and need to be incorporated once they are finalized. Additional refinements for local labor costs, equipment importation expenses, local distributor procurement mark-ups, and quality assurance programs would also improve accuracy.

The operational model will also impact effectiveness, as there is the potential for sample failure due to prolonged time between sample collection and laboratory receipt with the potential for cold-chain failure during shipment. This leads to trade-offs between a laboratory system design that relies on smaller labs with less opportunity to benefit from cost efficiencies, as compared with centralized labs that are lower cost but may introduce opportunities for sample failure due to longer shipment times from distributed sampling locations.

Of course, cost is not the only factor in a decision about which protocol to adopt. The approach that is ultimately selected must be 1) feasible in the local context (*e.g.*, trained staff and key reagents are available), 2) reliable despite possible logistical and laboratory delays (*e.g.*, shipping from remote locations), and 3) both sensitive and specific enough to meet surveillance needs. Thus, once the protocols have been laboratory- and field-validated, the costs will need to be balanced with effectiveness. Future work is needed to consider and assess the value of policy-relevant use cases for environmental surveillance. For example, some methods provide binary data on the presence versus absence of the pathogen, while others are quantified, and this information could have different levels of value for decision makers. Once these use cases are clarified, future studies can combine the cost estimates we provide here with a quantified value of information into a cost-effectiveness analysis. Combined with measures such as the lower limit of detection or the number of samples (or sites) required to meet a certain use case, this can be used to decide between protocols.

This study lays out the framework to investigate drivers of costs for environmental surveillance across a range of methodological approaches and protocols. Although variability exists between protocols' overall costs, there is a general observation that the economies of scale for environmental surveillance may be significant. With awareness of this dynamic, future

policymakers can use this information to design a surveillance and laboratory system that optimizes existing resources, and to consider how broadening surveillance may scale from a cost perspective.

## Supporting information

**S1 Text. Cost model input parameters.** Table A in S1 Text. Methods-based laboratory equipment requirements. Table B in S1 Text. Capacity of laboratory equipment. Table C in S1 Text. Unit costs for consumables. Table D in S1 Text. Operational paratmers for estimating labor-hours. Table E in S1 Text. Additional rates and values used in the cost model.
(DOCX)

**S2 Text. Survey on collection, concentration, and assay methods for environmental surveillance of S. Typhi.**
(DOCX)

**S3 Text. Total and operational costs per sample.** Table A in S3 Text. Total and operational costs per sample (modeled).
(DOCX)

**S4 Text. Cost model details.**
(DOCX)

## Acknowledgments

We would like to thank Dr. Supriya Kumar for her contributions to understanding the need for this study and what information would be most useful. We would also like to thank all of the members of the Typhoid environmental surveillance working group for their invaluable contributions and willingness to contribute unit cost and protocol information for use in this study.

## Author Contributions

**Conceptualization:** Brittany Hagedorn, Nicolette A. Zhou, Christine S. Fagnant-Sperati, Jeffry H. Shirai, Jillian Gauld, Yuke Wang, David S. Boyle, John Scott Meschke.

**Data curation:** Brittany Hagedorn, Nicolette A. Zhou, Christine S. Fagnant-Sperati, Yuke Wang.

**Formal analysis:** Brittany Hagedorn.

**Investigation:** Brittany Hagedorn.

**Methodology:** Brittany Hagedorn, Jillian Gauld, John Scott Meschke.

**Project administration:** Brittany Hagedorn, Jeffry H. Shirai.

**Resources:** John Scott Meschke.

**Supervision:** John Scott Meschke.

**Visualization:** Brittany Hagedorn, Jillian Gauld.

**Writing – original draft:** Brittany Hagedorn, Nicolette A. Zhou, Christine S. Fagnant-Sperati.

**Writing – review & editing:** Brittany Hagedorn, Nicolette A. Zhou, Christine S. Fagnant-Sperati, Jeffry H. Shirai, Jillian Gauld, Yuke Wang, David S. Boyle, John Scott Meschke.

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
