## [Decision Letter · Decision Letter 0]

24 Oct 2022

PGPH-D-22-01258

Estimates of the cost to build a stand-alone environmental surveillance system for typhoid in low- and middle-income countries

Dear Dr. Hagedorn,

Thank you for submitting your manuscript to PLOS Global Public Health. After careful consideration, we feel that it has merit but does not fully meet PLOS Global Public Health’s publication criteria as it currently stands. Therefore, we invite you to submit a revised version of the manuscript that addresses the points raised during the review process.

We look forward to receiving your revised manuscript.

Kind regards,

Habib Hasan Farooqui, MBBS, MD

Academic Editor

Journal Requirements:

1. We have amended your Competing Interest statement to comply with journal style. We kindly ask that you double check the statement and let us know if anything is incorrect.

a. Please clarify all sources of funding (financial or material support) for your study. List the grants (with grant number) or organizations (with url) that supported your study, including funding received from your institution. 

b. State the initials, alongside each funding source, of each author to receive each grant.

c. State what role the funders took in the study. If the funders had no role in your study, please state: “The funders had no role in study design, data collection and analysis, decision to publish, or preparation of the manuscript.”

d. If any authors received a salary from any of your funders, please state which authors and which funders.

3. Please provide the full and correct funding information for your study and confirm the order in which funding contributions should appear. 

4. We have noticed that you have uploaded Supporting Information files, but you have not included a list of legends. Please add a full list of legends for your Supporting Information files after the references list. 

5. In the online submission form you indicate that your data is not available for proprietary reasons and have provided a contact point for accessing this data. Please note that your current contact point is a co-author on this manuscript. According to our Data Policy, the contact point must not be an author on the manuscript and must be a third party. Please revise your data statement to a non-author institutional point of contact, such as a data access or ethics committee, and send this to us via return email. Please also include contact information for the third party organization, and please include the full citation of where the data can be found.

Additional Editor Comments (if provided):

Based on the analysis and results, given there is heterogeneity in the annualized cost across surveillance systems, especially with reference to labour, equipment and reagents - it would be helpful to provide clear guidance to the readers on approaches to setting up new surveillance systems under certain constraints such as high labour cost versus high equipment cost versus high recurring costs in LMICs settings. 

Reviewer 1 :

The manuscript gives a detailed estimation of the cost for thirteen protocols used in six countries. It is an important and timely study particularly when most countries are exploring investing in newer surveillance methods like environmental surveillance systems.

However, it will be more useful for policymakers in the LMICs if the authors can give recommendation on cost-effective protocol with respect to the sensitivity of the protocol. Future studies can also be recommended if needed.

It will add more value if authors can clarify on how the labor market dynamics from the LMICs are taken care of in calculating the average cost.

Reviewers' comments:

Reviewer's Responses to Questions

**Comments to the Author**

1. Does this manuscript meet PLOS Global Public Health’s publication criteria? Is the manuscript technically sound, and do the data support the conclusions? The manuscript must describe methodologically and ethically rigorous research with conclusions that are appropriately drawn based on the data presented.

Reviewer #1: Yes

2. Has the statistical analysis been performed appropriately and rigorously?

Reviewer #1: Yes

3. Have the authors made all data underlying the findings in their manuscript fully available (please refer to the Data Availability Statement at the start of the manuscript PDF file)?

Reviewer #1: Yes

4. Is the manuscript presented in an intelligible fashion and written in standard English?

Reviewer #1: Yes

5. Review Comments to the Author

Reviewer #1: The manuscript gives detailed estimation of cost for thirteen protocols used in six countries. It is an important and timely study particularly when most of the countries are exploring to invest in newer surveillance methods like environmental surveillance systems.

However, it will be more useful for policymakers in the LMICs if the authors can give recommendation on cost-effective protocol with respect to the sensitivity of the protocol. Future studies can also be recommended if needed.

It will add more value if authors can clarify on how the labor market dynamics from the LMICs are taken care of in calculating the average cost.

6. PLOS authors have the option to publish the peer review history of their article (what does this mean?). If published, this will include your full peer review and any attached files.

**Do you want your identity to be public for this peer review?** For information about this choice, including consent withdrawal, please see our Privacy Policy.

Reviewer #1: **Yes: **Dr. Shrikant Kalaskar

---

## [Editor Report · Decision Letter 1]

8 Dec 2022

Estimates of the cost to build a stand-alone environmental surveillance system for typhoid in low- and middle-income countries

PGPH-D-22-01258R1

Dear Ms. Hagedorn,

We are pleased to inform you that your manuscript 'Estimates of the cost to build a stand-alone environmental surveillance system for typhoid in low- and middle-income countries' has been provisionally accepted for publication in PLOS Global Public Health.

Best regards,

Habib Hasan Farooqui, MBBS, MD

Academic Editor